# Nontraditional Movement Behavior of Skyrmion in a Circular-Ring Nanotrack

**DOI:** 10.3390/nano13222977

**Published:** 2023-11-20

**Authors:** Na Cai, Xin Zhang, Yong Hu, Yan Liu

**Affiliations:** College of Sciences, Northeastern University, Shenyang 110819, China; 2010021@stu.neu.edu.cn (N.C.); 2200192@stu.neu.cn (X.Z.)

**Keywords:** skyrmion, circular-ring nanotrack, skyrmion Hall effect, Dzyaloshinskii–Moriya interaction

## Abstract

Magnetic skyrmions are considered promising candidates for use as information carriers in future spintronic devices. To achieve the development of skyrmion-based spintronic devices, a reasonable and feasible nanotrack is essential. In this paper, we conducted a study on the current-driven skyrmion movement in a circular-ring-shaped nanotrack. Our results suggest that the asymmetry of the inside and outside boundary of the circular ring changed the stable position of the skyrmion, causing it to move like the skyrmion Hall effect when driven by currents. Moreover, the asymmetric boundaries have advantages in enhancing or weakening the skyrmion Hall effect. Additionally, we also compared the skyrmion Hall effect from the asymmetric boundary of circular-ring nanotracks with that from the inhomogeneous Dzyaloshinskii–Moriya interaction. It was found that the skyrmion Hall effect in the circular ring is significantly greater than that caused by the inhomogeneous Dzyaloshinskii–Moriya interaction. These results contribute to our understanding of the skyrmion dynamics in confined geometries and offer an alternative method for controlling the skyrmion Hall effect of skyrmion-based devices.

## 1. Introduction

Skyrmions are fascinating particle-like swirling spin configurations that have garnered significant attention since their inception [1,2]. Their topological properties make them incredibly stable and endow them with exotic particle dynamics behavior [3,4,5]. Moreover, the successful experimental observation and controllability of skyrmions in a wide temperature range and near-zero field opened the door for their application in spintronic devices [2,6,7,8,9,10]. Due to the topological stability, low driving current, and small size, skyrmions are expected to further enhance the performance of spin electronic devices [11,12,13,14]. In other words, the development of skyrmion-based devices is expected to improve the current information storage capacity of devices.

It is well known that when driven by the current, the skyrmion Hall effect (SkHE) causes skyrmions to drift away from the driving current direction [15,16,17]. The processing of SkHE is one crucial factor in advancing skyrmion-based devices. In the past, the SkHE was viewed as a harmful problem for devices, and great efforts were made to suppress it [18,19]. However, recent research has shown that the SkHE can be utilized to design various devices [16,20,21,22,23,24,25,26,27,28,29]. For example, Hong et al. take advantage of skyrmions’ large transverse motion induced by SkHE-designed transistors [20]. Whang et al. proposed a fast and robust oscillator that utilizes the SkHE to modulate the skyrmion’s movement [27]. Additionally, several diodes have been proposed and implemented using the SkHE-induced unidirectional transportation of skyrmions [24,25,26,29]. In other words, the SkHE can control the movement of the skyrmion when it performs different functions in devices.

Apart from controlling the SkHE, the effect of nanotrack boundaries on the skyrmion movement in skyrmion-based devices cannot be disregarded. Studies have indicated that the nanotrack’s boundaries have a significant impact on the behavior of skyrmions [24,30,31,32,33,34,35,36,37,38,39,40]. As reported by Leonov et al., the edge state not only repulses skyrmions, but also attracts them and guides their movement [36,37]. Additionally, Saidi et al. observed that skyrmions shrink in size and follow a spiral path as they approach the edge of the track [38]. Another study conducted by Morshed’s team revealed that the boundary with notches has a significant pinning effect on skyrmions [39]. Inspired by this motivation, we investigate the current-driven behavior of skyrmions in circular-ring nanotracks. Compared with the symmetrical straight nanotrack, the inner and outer boundaries of the circular ring are asymmetric, which may induce some special phenomena.

## 2. Materials and Methods

We modeled ultra-thin ferromagnetic Co/Pt with a circular-ring nanotrack, as shown in Figure 1a, which can be experimentally fabricated using lithography techniques [41,42]. Our simulations were carried out using the mumax3 software package (version 3.6) [43,44]. The circular-ring nanotrack was parametrized as $\gamma_{O}=R_{m}\cdot (e_{x}cos\theta +e_{y}\sin\theta )$, where *R***_m_** is the middle line radius of the track and (***e***_x_, ***e***_y_) is the Cartesian basis. The simulation mesh was divided into 110 × 110 × 1 grids with a cell size of 2 × 2 × *h* nm^3^, where *h* represents the thickness of the track. The choice of cell size was much lower than the exchange length *l*_ex_ and skyrmion size (approximately 13.5 nm in our simulation). To maintain accuracy, we chose a 2 nm cell size, which was also used for Co/Pt in other literature [32,38,39]. Under zero field, skyrmions with two polarities are stabilized at the initial center of the track because of the uniaxial anisotropy. There is a certain range of *K*_u_ values that can ensure the stable injection of an isolated skyrmion [45]. In our system, only when $6.1\times 10^{5}\;J\cdot m^{-3}\leq K_{u}\leq 9.2\times 10^{5}\;J\cdot m^{-3}$ can we obtain a stable circular skyrmion, while when $4.4\times 10^{5}\;J\cdot m^{-3}<K_{u}<6.2\times 10^{5}\;J\cdot m^{-3}$ is an elliptical instability state and when $K_{u}\leq 4.4\times 10^{5}\;J\cdot m^{-3}$, irregular magnetic domains or helical vortex states appear. In experiments, skyrmion can be produced through electric pulse, magnetic field, and other methods [46,47]. Then, we injected an in-plane current to drive the skyrmion moving along the circular-ring nanotrack. The red arrows in Figure 1a represent the clockwise (CW) and counterclockwise (CCW) directions of the current, which were both equally dispersed across the circular ring. We refer to the situation of the skyrmion traveling clockwise in the full text as CW, and the case of the skyrmion traveling counterclockwise as CCW.

The magnetization dynamics driven by an in-plane current are governed by the Landau–Lifshitz–Gilbert equation containing the spin-transfer torque (STT) effect:(1)$$\dot{m}=-\gamma m\times H_{eff}+\alpha \left(m\times \dot{m}\right)+m\times \left[m\times (u\cdot \nabla )m\right]+\beta m\times (u\cdot \nabla )m.$$
where γ is the gyromagnetic ratio, **m** = M/*M*_s_ is the normalized magnetization vector, and *M*_s_ is the saturation magnetization. The effective magnetic field is given by $H_{eff}=-(\delta E/\delta m)/\mu_{0}M_{s}$, where $\mu_{0}$ is the vacuum permeability, and *E* is the magnetic energy of the system, including energy terms such as exchange, anisotropy, Dzyaloshinskii–Moriya interaction (DMI), Zeeman, and demagnetization energy. The interfacial DMI energy density is given by $E_{\mathrm{DM}}=-D_{M}m\cdot ((\hat{z}\times \nabla )\times m)$, where *D*_M_ is the DMI constant. For $\mu =\mu_{B}jp/eM_{s}$, *j* is the current density, *p* is the spin polarization rate, $\mu_{B}$ is the Bohr magneton, and ***e*** is the electron charge. α and β are the Gilbert damping constant and the non-adiabaticity factor, respectively. We chose the case α=β=0.01 here to exclude the intrinsic SkHE [15,48]. The magnetic parameters we used here are as follows: $M_{s}=5.8\times 10^{5}\;A\cdot m^{-1}$, $A=15\times 10^{-12}\;J\cdot m^{-1}$, $D_{M}=3\times 10^{-3}\;J\cdot m^{-2}$, $K_{u}=8\times 10^{5}\;J\cdot m^{-3}$, *p* = 0.5.

## 3. Results

The functions between skyrmion velocity and the position in the circular-ring nanotrack enabled us to determine the dynamics of the skyrmion. We calculated the skyrmion velocity and decomposed it into two components, parallel velocity *V*_T_ and perpendicular velocity *V*_N_, where *V*_T_ (*V*_N_) is the velocity component along (perpendicular to) the current direction. The positive directions of *V*_T_ and *V*_N_ are along $e_{T}$ and $e_{N}$, where $e_{T}$ is the tangent direction of the circular ring, and $e_{N}$ is the normal direction of the circular ring. It can be seen from Figure 1b that *V*_T_ shows a periodic trend of increasing first and then decreasing in both CCW and CW directions, where the *V*_T_ of CCW (CW) with *Q* = −1 overlaps with CW(CCW) with *Q* = 1. This fluctuation also occurs in *V*_N_, as shown in Figure 1c, which fluctuates over a non-zero value and has the same periodicity as *V*_T_. This fluctuation of velocity is significantly different from the case in a straight track. In a straight track, the velocity remains constant in parallel and zero in the perpendicular direction for α=β. However, in the circular-ring nanotrack, the velocity fluctuates significantly. To compare the movement behavior of a skyrmion in the straight and circular-ring tracks, we define *R***_sk_** as the radius of the real moving trajectory and compare it with the middle line radius of the circular ring *R***_m_**. Here, we let *R***_d_** = *R***_m_** − *R***_sk_**, and *R***_d_** is the distance between the middle line and the skyrmion’s real moving trajectory. Figure 1d shows the variation in *R***_d_** with the angle position. The initial non-zero value of *R***_d_** indicates that the skyrmion undergoes a transverse drift from the middle line. This means that the skyrmion first stabilizes inside the middle line and then moves towards the inner or outer boundary driven by the current.

Upon the observation of skyrmion movement in the circular-ring nanotrack, we found that the skyrmion moves at a certain angle to the driving current, similar to the movement caused by the skyrmion Hall effect (SkHE). We calculated the Hall angle $\theta_{\mathrm{SkHE}}$ in the circular-ring nanotrack and show it in Figure 2a, where $\theta_{\mathrm{SkHE}}=\arctan(V_{N}/V_{T})$. It is obvious that the Hall angle is non-zero, and the maximum value is about 2°. We found that this type of SkHE still exists for different track thicknesses (*h* = 0.4 nm and *h* = 1 nm). In traditional SkHE, the key influencing factors are the skyrmion movement direction and the topological number *Q*, as shown in the gray section of Figure 2a. Similar to the traditional SkHE, the Hall angle in the circular-ring nanotrack also demonstrates a notable reliance on both the skyrmion movement direction and the topological number *Q*. For a skyrmion with *Q* = −1, the CCW current drives it towards the inner side, while the CW current drives it towards the outer side of the circular-ring nanotrack. The situation is exactly the opposite for a skyrmion with *Q* = 1. 

Based on the obtained results, it can be concluded that the movement of skyrmions in circular-ring nanotracks is similar to the SkHE. However, the SkHE does not occur in the straight nanotrack when α=β. This leads us to assume that the SkHE observed in circular rings may be due to the asymmetric inner and outer boundaries of the circular ring. In Figure 2b, we show the function of *V*_T_ and the drift distance *R***_d_** for different *R***_m_**. It can be found that *V*_T_ is linearly related to *R***_d_** in both movement directions: it is just a negative correlation in CW and a positive correlation in CCW. The situations when *Q* = 1 are the opposite. Furthermore, special attention should be paid to the fact that the stable positions of the skyrmion, represented by the intersection of the velocities in two directions, vary significantly for circular-ring nanotracks of different sizes. The size of the circular-ring nanotrack has a direct impact on the stable position of the skyrmion. As the size of the ring decreases, its curvature becomes more prominent, resulting in less symmetry between its inner and outer boundaries. Conversely, larger circular rings appear more symmetrical, with smaller differences between the inner and outer boundaries. Due to the asymmetric boundary of the circular-ring nanotrack, the stable position of the skyrmion drifts farther from the middle line as the circular-ring nanotrack size decreases. As a result, the *R***_d_** (initial position) in Figure 1d is not zero. When a skyrmion moves along a straight track, it remains stable on the middle line without deviation. However, when a skyrmion moves along a circular-ring track, the collision between the skyrmion and the asymmetric boundaries results in a periodic change in the skyrmion’s Hall angle. This, in turn, leads to periodic increases and decreases in skyrmion velocity. Therefore, the period of such collisions is uncertain. When we modify the size of the nanotrack or the driving current density, the SkHE still exhibits an oscillation mode, but the period changes accordingly. In topological memory with half-vortices pairs and half-antivortices pairs as information storage carriers, the ring-shaped boundary also plays an important role in its functioning. According to the research by Metlov, the use of multiple connected elements constructed by the ring-shaped boundary can greatly increase the information storage capacity of the topological memory [49]. 

To obtain a deeper understanding of how asymmetric boundaries affect the movement of skyrmions in circular-ring nanotracks, the Thiele equation is utilized. Since the size of skyrmion changes is not obvious and the topological structure is maintained, we still treat the skyrmion as a rigid object in the asymmetric track. This rigid approximation allows the Thiele equation for the skyrmion to be written as:(2)$$G\times (V^{e}-V^{\mathrm{sk}})+D(\beta V^{e}-\alpha V^{\mathrm{sk}})-\nabla U=0,$$
where the first term is the Magnus force, ***G*** is the gyrovector related to the topological number *Q* of skyrmion, and ***G*** = (0, 0, 4π*Q*) is a constant for the Néel skyrmion. The second term of Equation (2) stands for the dissipative force and the *D* comes from a dimensionless matrix related to the dissipative force, being its components *D_xx_* = *D_yy_* = *D*. $V^{e}=(V^{e},0)=(-Pl^{3}j/2eM_{s},0)$ is the velocity of the electron, and *l* is the lattice constant of the material. **V**^sk^ = (*V*_T_, *V*_N_) is the skyrmion velocity that can be decomposed into two orthogonal parts (the tangential $e_{T}$ and normal $e_{N}$ directions of the circular-ring nanotrack). The third term, −∇U, represents the force **F** acting on the moving skyrmion and **F** = (*F*_T_, *F*_N_), including the forces from the boundary and driving current. 

For α=β, we can obtain the simplification form of the skyrmion velocity from Equation (2):(3)$$V^{\mathrm{sk}}=\left(\begin{array}{c} V_{T}\\ V_{N} \end{array}\right)=\frac{1}{G^{2}+\alpha^{2}D^{2}}\left(\begin{array}{c} (G^{2}+\alpha^{2}D^{2})V^{e}+\alpha DF_{T}+GF_{N}\\ -GF_{T}+\alpha DF_{N} \end{array}\right)$$

Considering *D* = |*G*| and $\alpha \ll 1\;(\alpha^{2}+1\approx 1)$, the function of velocity can be further simplified as:(4)$$\begin{array}{l} V_{T}=V^{e}+\frac{\alpha F_{T}}{D}-\frac{F_{N}}{G}\\ V_{N}=\frac{\alpha F_{N}}{D}+\frac{F_{T}}{G}\; \end{array}$$

It can be concluded from Equation (4) that the *V*_T_ is determined by the sign of $V^{e}$ and the product of *G* (*Q*) and *F*_N_. The force *F*_N_ acting on the skyrmion mainly comes from the repulsive force from the boundary. When a skyrmion moves towards the inner boundary, the force *F*_N_ is negative. On the other hand, when a skyrmion moves towards the outer boundary, the force *F*_N_ is positive. For a skyrmion with *Q* = −1, when it moves counterclockwise and approaches the inner boundary, its velocity increases. However, when it approaches the outer boundary, its velocity decelerates. For skyrmions with *Q* = 1, the situation is the opposite, since *G* is positive. This is consistent with the variations in skyrmion velocity described in Figure 1b.

The SkHE caused by the asymmetry of the inner and outer boundaries is also reflected by the drift in the skyrmion’s stable position. To study this relationship, we calculated the initial drift distance *R***_d_** for different sizes of *R***_m_** of the circular-ring nanotrack under a range of values α(β). The results presented in Figure 3a indicate that the skyrmion’s stable position tends to drift further away from the ring’s middle line as the circular ring becomes smaller. It can be observed from the inset of Figure 3a that the *R***_d_** is directly proportional to the curvature $\kappa =R_{m}^{-1}$ of the nanotrack, despite the values of α(β) being different. This is because the symmetry of the boundary is directly affected by the curvature of the circular ring. Additionally, both Figure 3a and the illustration in it show that the dependence of the skyrmion’s size $\phi_{\mathrm{sk}}$ (the red line) on the *R***_m_** is the same as that of *R***_d_**. The force from the asymmetric border essentially changes the size of the skyrmion and, hence, changes its stable position on the track.

In addition to the size *R***_m_** of the circular-ring nanotrack, the width also affects the force exerted by the boundary [40,50]. Figure 3b illustrates the drift distance *R***_d_** changes with the track’s width when the *R***_m_** is fixed at 80 nm, 100 nm, and 120 nm. Generally, it is still true that the smaller the *R***_m,_** the more obvious the skyrmion’s stable state drift from the middle line. However, for the same *R***_m_** values, the stable state can be divided into four regions at different width values, as shown in Figure 3b. In region I (0 < *W* < 40 nm), the tracks are too narrow for a stable skyrmion. In region II (40 < *W* < 70 nm), the presence of an asymmetric boundary leads to a reduction in the skyrmion’s size. However, the increase in width significantly increases the size of the skyrmion, causing *R***_d_** to gradually increase until the increase in width has little effect on the size of the skyrmion. When *W* > 70 nm, the skyrmion size does not increase significantly, so the *R***_d_** in regions III and IV is consistent with the effect of the circular ring size. For the larger *W* in region IV (*W* > 140 nm), the corresponding *R***_d_** ≤ 0.15 nm and the drift induced by the asymmetric boundary can be almost ignored. This is because as the track becomes wider, although the force caused by the asymmetric boundary still exists because the boundary is too far away from the skyrmion, the stable position will hardly move.

In order to confirm our analysis, we also calculated the skyrmion velocity on a wider nanotrack. Our findings revealed that as the track width widens, the periodic oscillation of *V*_T_ (shown in Figure 1b) diminishes or disappears entirely, and the *V*_N_ (depicted in Figure 1c) comes close to zero. To verify this, we tested a track width of *W* = 150 nm. As shown in Figure 4a,b, the *V*_T_ no longer fluctuates significantly, but remains close to the velocity in the long straight track, and *V*_N_ is also close to zero, as expected. The reason for this is that the boundary’s symmetry remains fixed for a given *R***_m_**, but the repulsive force from the boundary weakens as the track widens. This force is related to the skyrmion size and the distance from the boundary. As a result, the skyrmion is almost unaffected by the boundary.

Since the asymmetric boundaries of a circular ring can generate the SkHE, we have also studied the impact of asymmetric boundaries on the circular rings that already have the SkHE. As illustrated in Figure 4c,d, when α>β, the Hall angle in a circular-ring nanotrack is greater than that in a straight track, and the SkHE is enhanced. On the other hand, when α<β, the Hall angle in the circular-ring nanotrack is smaller than that in a straight track, and the SkHE is weakened. Therefore, the asymmetric boundaries can be used to strengthen or weaken the SkHE. This means that utilizing a circular-ring nanotrack offers a promising way to manipulate the SkHE in devices. 

According to the research, the SkHE can also be induced by the inhomogeneous DMI [51]. In this section, we compared the effect of the asymmetric boundary with that of the inhomogeneous DMI. We created straight nanotracks with an inhomogeneous DMI, which had the same width as the circular-ring nanotrack. First, we compared the effect of non-uniform DMI on the skyrmion’s stable position with the circular ring. Figure 5a illustrates the relationship between the drifted distance of the skyrmion’s stable position and the DMI change values for different initial DMI values. When there was no change in DMI, the skyrmion stayed in the center of the track. However, even a slight change in DMI can cause the skyrmion to drift from the middle line. The $1.25\times 10^{-6}\;J\cdot m^{-2}/\mathrm{nm}$ DMI gradient can cause a similar drift distance of the skyrmion’s stable position with the circular ring with *R***_m_** = 80 nm.

We also conducted an analysis to determine the factors that influence the Hall angle in tracks with the inhomogeneous DMI distribution. The trajectory of the skyrmion in the modeled nanotrack is shown in Figure 5b. It was observed that the presence of an inhomogeneous DMI distribution in the nanotrack causes the skyrmion to move toward the boundaries as it moves in the circular ring. Additionally, we found that the sign of the Hall angle depends on the sign of $\delta_{D_{M}}$ and skyrmion topological number *Q*. Figure 5c illustrates the relationship between the Hall angle and the value of the DMI changes for various current densities. Compared with the circular ring, the $1.1\times 10^{-7}\;J\cdot m^{-2}/\mathrm{nm}$ DMI gradient can only yield a Hall angle of about 2° for the $j=3.0\times 10^{-3}\;A\cdot m^{-2}$ driving current. For high current density, a larger DMI gradient is required to cause a 2° Hall angle. Therefore, the SkHE caused by the circular ring is much more significant than that caused by the inhomogeneous DMI.

## 4. Conclusions

In conclusion, we investigated the current-driven skyrmion movement in circular-ring-shaped nanotracks. The results showed that the presence of an asymmetric boundary causes a shift in the skyrmion’s stable position, leading to an SkHE-like movement when driven by a current. We also noticed that the skyrmion velocity can be linearly changed by the SkHE induced by the asymmetric boundary of the circular-ring nanotrack. The asymmetric boundaries also have advantages in enhancing or weakening the skyrmion Hall effect. We also compared the SkHE from the asymmetric boundaries of circular-ring nanotracks with that from the inhomogeneous DMI. It was found that the SkHE caused by the circular ring is significantly greater than that caused by the inhomogeneous DMI. These findings not only contribute to our understanding of the skyrmion dynamics in confined geometries, but also provide an alternative approach to controlling the SkHE of skyrmion-based devices. Due to the significant impact of curved tracks on skyrmion motion, more interesting movement behaviors of single or multiple skyrmions can be expected in different shaped tracks such as disks, Corbino geometric rings, coupling rings, and elliptical rings. Changing the shape of the track is also expected to provide the possibility of the manipulation of the skyrmion velocity in devices.

## Figures and Tables

**Figure 1 nanomaterials-13-02977-f001:**
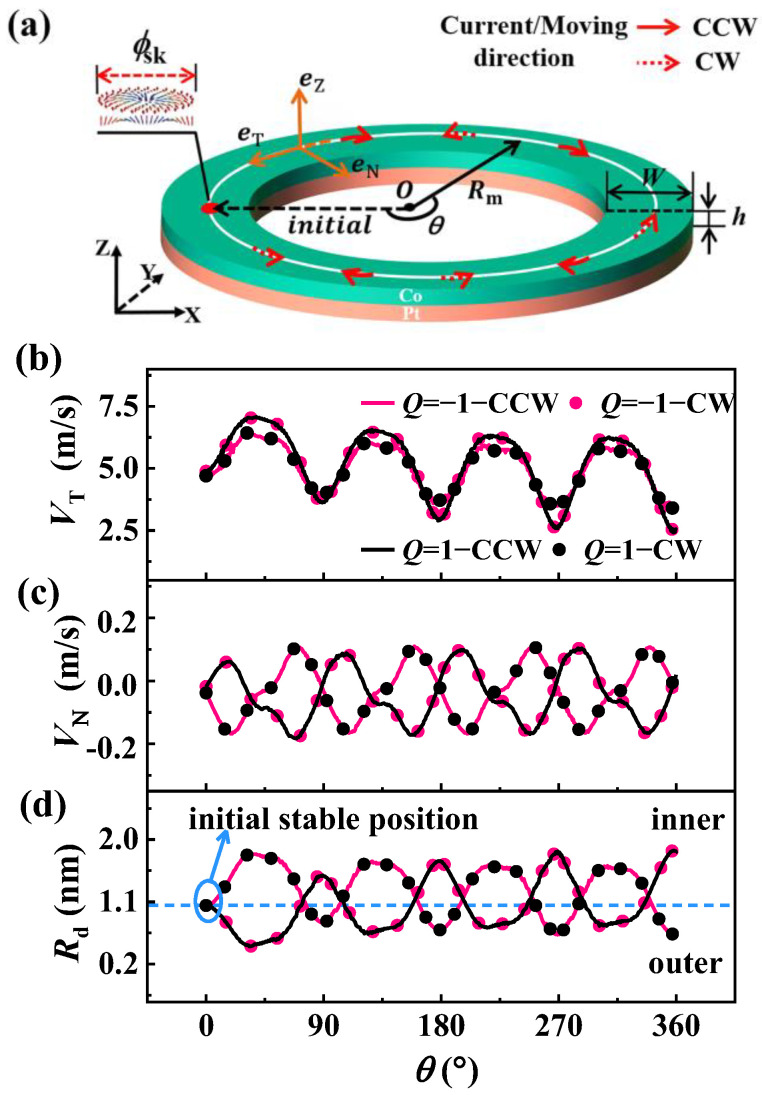
Model and skyrmion dynamics. (**a**) The schematic of the circular-ring nanotrack, where *θ* represents the angular parameter of the skyrmion at different positions, and *R***_m_** is the middle line radius of the tracks. The red arrows represent the clockwise (CCW) and counterclockwise (CW) driving current (moving) directions, respectively. (**b**) The parallel velocity *V*_T_, perpendicular velocity *V*_N_ (**c**), and the drift distance of the initial stable position (**d**) as functions of the position parameter *θ*. *Q* represents the topological number of skyrmion.

**Figure 2 nanomaterials-13-02977-f002:**
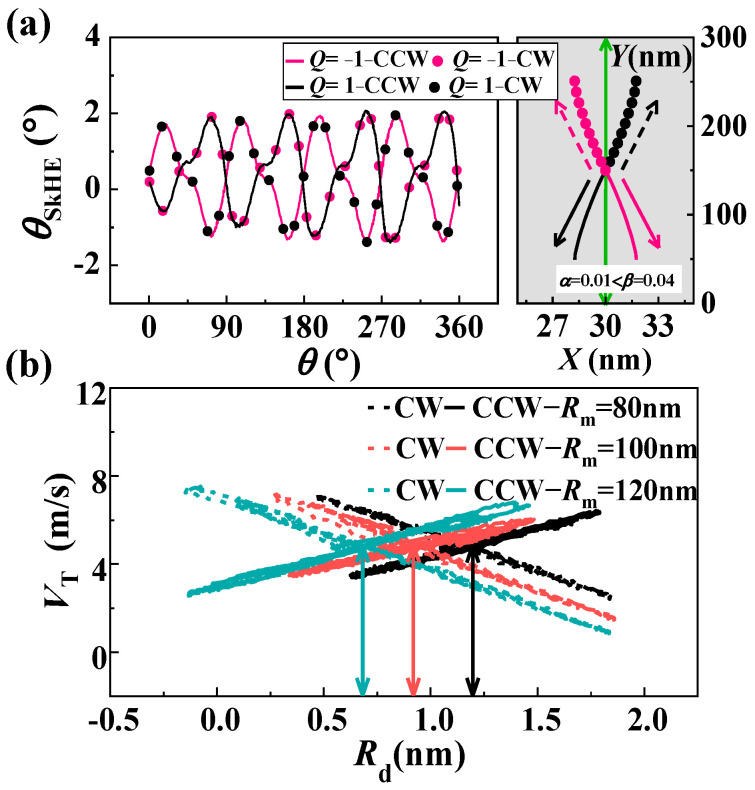
Skyrmion dynamics in the circular ring. (**a**) The skyrmion Hall angle as a function of the position parameter *θ*. The gray part shows the trajectories of the skyrmion under the conventional skyrmion Hall effect for the case α<β. (**b**) The function between *V*_T_ and the drift distance *R***_d_** for different middle line radius *R***_m_** values. Unless otherwise specified, the simulations were carried out under the fixed width *W* = 60 nm and the applied current density $j=1\times 10^{11}\;A\cdot m^{-2}$.

**Figure 3 nanomaterials-13-02977-f003:**
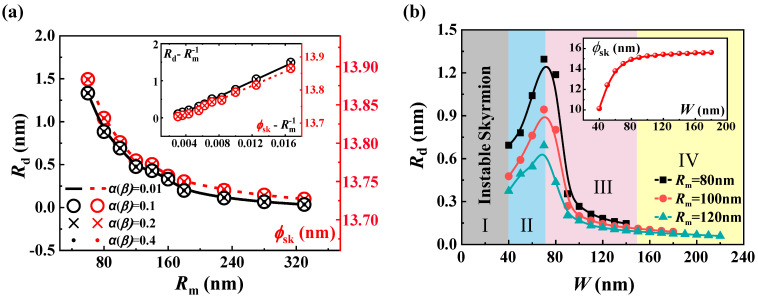
Analysis of skyrmion dynamics. (**a**) The drift distance *R***_d_** (black line) and the size of the skyrmion $\phi_{\mathrm{sk}}$ (red line) as functions of the middle line radius *R***_m_** for the cases of no current and different α(β) values. (**b**) The drift distance *R***_d_** as a function of track width *W*, where the *R***_m_** is fixed at 80 nm, 100 nm, and 120 nm, respectively. Unstable region I, width-dominant region II, asymmetric-boundary-dominant region III, low response region IV.

**Figure 4 nanomaterials-13-02977-f004:**
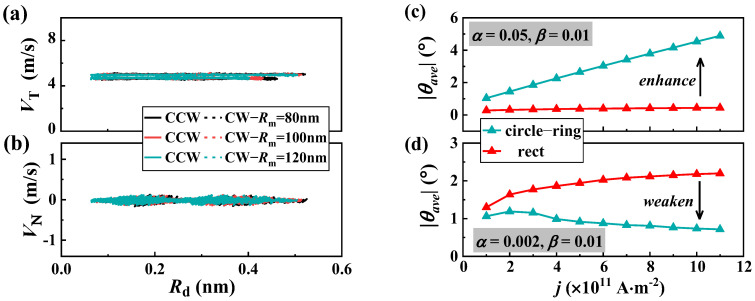
Verification of the skyrmion dynamics. The skyrmion velocity’s (**a**) parallel velocity *V*_T_ and (**b**) perpendicular velocity *V*_N_ as functions of its position parameter *θ* with different middle line radius *R***_m_**; the width is fixed at 150 nm. The average values of the Hall angle as functions of the current density in circular-ring nanotrack and straight track for the cases of (**c**) α>β and (**d**) α<β.

**Figure 5 nanomaterials-13-02977-f005:**
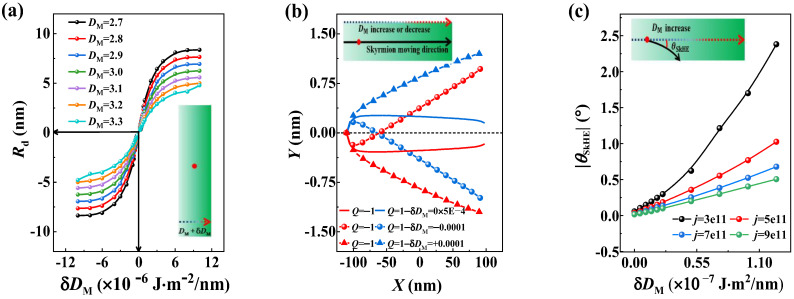
Skyrmion dynamics in an inhomogeneous DMI distribution nanotrack. (**a**) The drift distance *R***_d_** as a function of the DMI change value in long straight tracks for different initial DMI values. (**b**) Skyrmion motion trajectory driven by the current in the modeled nanotrack, with the modeled track with an inhomogeneous DMI distribution in the long direction. (**c**) The Hall angle as a function of the DMI change value in the modeled nanotrack.

## Data Availability

All of the data present in this paper will be made available upon reasonable request. Please contact the corresponding author for further information.

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
