# Peer review of "Nontraditional Movement Behavior of Skyrmion in a Circular-Ring Nanotrack"

_nanomaterials, 2023, doi:10.3390/nano13222977_

Round 1

Reviewer 1 Report

Comments and Suggestions for Authors

The manuscript provides a detailed theoretical analysis of current-driven skyrmion movement in circular ring-shaped nanotracks. The main emphasis is put on the skyrmion Hall effect induced by the asymmetry of the ring boundaries. This is a novel result important for controlling the skyrmion dynamics. The manuscript is clearly written and well structured. Therefore, I would recommend this manuscript for publication in the Nanomaterials journal.

Note, however, that a similar problem, but from another viewpoint, was treated in the recent paper K.L. Metlov, JETP Letters 118 (2), 105–111 (2023). It would be interesting to compare the results of both approaches. It would be also reasonable to give a reference to the seminal paper on the racetrack memory: S.S.P. Parkin, M. Hayashi, and L. Thomas, Science 320, 190 (2008).

Reviewer 2 Report

Comments and Suggestions for Authors

In the present manuscript, the authors investigate the current-driven movement of skyrmions in circular-ring nanotracks. They demonstrate the asymmetry of such dynamics related to inner and outer boundaries.

In principle, the manuscript is clear enough and might be of potential interest for skyrmionic community. I therefore think that the manuscript might be suitable for publication.

Still, I have some recommendations to improve the manuscript:

1.      In the abstract, the authors write “As an outstanding information carrier, skyrmion is increasingly widely used in devices based on complex geometries.” To the best of my knowledge, skyrmions are used in spintronic devices only in numerical simulations. It is still quite far from their practical realization. Therefore, I think the authors should be careful with such a sort of statements.

2.      I would advise to extend the introduction about the influence on skyrmions of so-called edge states formed at the boundaries of nanomagnets. It is known that such edge state repulse skyrmions, but they can also attract them. Here are two references on the edge states:

Appl. Phys. Lett. 109, 172404 (2016);

Nat. Commun. 8, 14394 (2017)

3.      In section 2, I would ask the authors to provide more technical details on the computational procedures. For example, what is the grid size? Why the authors use the cell size 2 nm instead of 1 nm? How is the cell size comparable with the sizes of skyrmions?

4.      In the same section 2, I think, the authors should explicitly write that the field is zero and skyrmions with two polarities are stabilized because of the uniaxial anisotropy. The authors may also calculate the non-dimensional anisotropy ku to show how far are they from the region of the helical state which would induce the elliptical instability of isolated skyrmions. The corresponding phase diagram can be found in Ref. [New J. of Phys. 18, 065003 (2016)].

5.      The authors should introduce the vectors eT and eN.

6.      It is not clear why is there any difference between CW and CCW skyrmion motion? Why just one case of CW motion cannot be considered for skyrmions with two polarities?

7.      Fig. 1(a) is confusing. Are the black and red arrows needed? On the contrary, the directions seem to be interchanged.

8.      In Fig. 1 (c) and 2 (a) the meaning of dots and lines is not clear. Indeed, the pink line signifies the CCW motion of Q=-1 skyrmions, but the pink points are for the skyrmions with the opposite charge.

9.      Why is the initial position in Fig. 1 (d) not zero?

10.  Would it be possible to show the trajectory of moving skyrmions directly on a ring? This would be much more transparent. Probably, some animations could be prepared too?

11.  The authors claim that there is no dependence on the thickness. But since the grid size along z is probably just 1, then, by changing h, one just changes the discretization of derivatives. This statement should be elaborated.

Reviewer 3 Report

Comments and Suggestions for Authors

Report for

Abnormal Movement Behavior of Skyrmion in a Circular-ring 2
Nanotrack

N. Cai et al,

This paper considers the motion of skyrmion in a ring-like nanotrack with asymmetry in the boundaries. The authors consider micromagnetic simulations
of skyrmion in a ring. They see a skyrmion Hall angle
and map the effect for clockwise and counterclockwise currents, changing DMI on the track and width. An interesting effect is that the skyrmion Hall
angle can be reduced in some cases, which would be useful for applications. The authors also show that several results can be understood with the Thiel equation approach. There has been considerable work on skyrmion in straight channels, so the ring-like systems have yet to be studied but are certainly experimentally feasible, so the results are new. The paper also fits in well with this special issue. The paper could be published after the modifications mentioned below.

(1) The authors mention diodes, directed transport, and ratchets where the skyrmion Hall effect is important when some kind of asymmetry is present but left out
a series of essential citations where the Hall effect on ratchets and diodes was explicitly considered.

 Magnus induced diode effect for skyrmions in channels with periodic potentials",
J.C. Bellizotti Souza, N.P. Vizarim, C.J.O. Reichhardt, C. Reichhardt and P.A. Venegas
J. Phys.: Condens. Matter 51 015804 (2023).

Clogging, diode and collective effects of skyrmions in funnel geometries",
J.C. Bellizotti Souza, N.P. Vizarim, C.J.O. Reichhardt, C. Reichhardt and P.A. Venegas
New J. Phys. 24 , 103030 (2022).

(2) A future direction is considering a  disk or gradient in the current, like the Corbino geometry. The authors could mention this as a future direction; other could be coupled rings or multiple skyrmions.

(3) The overall grammar could be improved; the paper is readable, but the wording is awkward

For example
"As an outstanding information carrier, skyrmion is increasingly widely used in devices 7
based on complex geometries."

It could be better worded.

"As outstanding information carriers, skyrmions are
increasingly being used widely in devices 7
with complex geometries.

"Skyrmion is a fascinating particle-like swirling spin configuration that has garnered 23
significant attention since its inception [1, 2]. "

Could be
"Skyrmions arefascinating particle-like swirling spin configurations
 that ahve garnered 23
significant attention since their inception [1, 2]. "

These are too numerous for me to go through individually, but the authors should
look carefully through the paper.

(4) The title is catchy, but I am unsure if the word Abnormal
is correct. One could have "Controlled" or simply "Motion of skyrmions in
Circular-ring Nantracks.
It is up to the authors if they wish to change this.

Comments on the Quality of English Language

Report for

Abnormal Movement Behavior of Skyrmion in a Circular-ring 2
Nanotrack

N. Cai et al,

This paper considers the motion of skyrmion in a ring-like nanotrack with asymmetry in the boundaries. The authors consider micromagnetic simulations
of skyrmion in a ring. They see a skyrmion Hall angle
and map the effect for clockwise and counterclockwise currents, changing DMI on the track and width. An interesting effect is that the skyrmion Hall
angle can be reduced in some cases, which would be useful for applications. The authors also show that several results can be understood with the Thiel equation approach. There has been considerable work on skyrmion in straight channels, so the ring-like systems have yet to be studied but are certainly experimentally feasible, so the results are new. The paper also fits in well with this special issue. The paper could be published after the modifications mentioned below.

(1) The authors mention diodes, directed transport, and ratchets where the skyrmion Hall effect is important when some kind of asymmetry is present but left out
a series of essential citations where the Hall effect on ratchets and diodes was explicitly considered.

 Magnus induced diode effect for skyrmions in channels with periodic potentials",
J.C. Bellizotti Souza, N.P. Vizarim, C.J.O. Reichhardt, C. Reichhardt and P.A. Venegas
J. Phys.: Condens. Matter 51 015804 (2023).

Clogging, diode and collective effects of skyrmions in funnel geometries",
J.C. Bellizotti Souza, N.P. Vizarim, C.J.O. Reichhardt, C. Reichhardt and P.A. Venegas
New J. Phys. 24 , 103030 (2022).

 "Skyrmion ratchet in funnel geometries",
J.C. Bellizotti Souza, N.P. Vizarim, C.J.O. Reichhardt, C. Reichhardt and P.A.Venagas
Phys. Rev. B 104 , 054434 (2021).

 "Guided skyrmion motion along pinning array interfaces",
N.P. Vizarim, C. Reichhardt, P.A. Venegas, and C.J.O Reichhardt
J. Magn. Magn. Mater 528 167710 (2021).

(2) A future direction is considering a  disk or gradient in the current, like the Corbino geometry. The authors could mention this as a future direction; other could be coupled rings or multiple skyrmions.

(3) The overall grammar could be improved; the paper is readable, but the wording is awkward

For example
"As an outstanding information carrier, skyrmion is increasingly widely used in devices 7
based on complex geometries."

It could be better worded.

"As outstanding information carriers, skyrmions are
increasingly being used widely in devices 7
with complex geometries.

"Skyrmion is a fascinating particle-like swirling spin configuration that has garnered 23
significant attention since its inception [1, 2]. "

Could be
"Skyrmions arefascinating particle-like swirling spin configurations
 that ahve garnered 23
significant attention since their inception [1, 2]. "

These are too numerous for me to go through individually, but the authors should
look carefully through the paper.

(4) The title is catchy, but I am unsure if the word Abnormal
is correct. One could have "Controlled" or simply "Motion of skyrmions in
Circular-ring Nantracks.
It is up to the authors if they wish to change this.

Round 2

Reviewer 2 Report

Comments and Suggestions for Authors

The authors addressed all my concerns.